# WEAKLY SUPERVISED NEURO-SYMBOLIC MODULE NETWORKS FOR NUMERICAL REASONING

## ABSTRACT

Neural Module Networks (NMNs) have been quite successful in incorporating explicit reasoning as learnable modules in various question answering tasks, including the most generic form of numerical reasoning over text in Machine Reading Comprehension (MRC). However, to achieve this, contemporary NMNs need strong supervision in executing the query as a specialized program over reasoning modules and fail to generalize to more open-ended settings without such supervision. Hence we propose **W**eakly-**S**upervised **N**euro-**S**ymbolic **M**odule **N**etwork (WNSMN) trained with answers as the sole supervision for numerical reasoning based MRC. It learns to execute a noisy heuristic program obtained from the dependency parsing of the query, as discrete actions over both neural and symbolic reasoning modules and trains it end-to-end in a reinforcement learning framework with discrete reward from answer matching. On the numerical-answer subset of DROP, WNSMN outperforms NMN by 32% and the reasoning-free language model GenBERT by 8% in exact match accuracy when trained under comparable weak supervised settings. This showcases the effectiveness and generalizability of modular networks that can handle explicit discrete reasoning over noisy programs in an end-to-end manner.

## 1 INTRODUCTION

End-to-end neural models have proven to be powerful tools for an expansive set of language and vision problems by effectively emulating the *input-output* behavior. However, many real problems like Question Answering (QA) or Dialog need more interpretable models that can incorporate explicit reasoning in the inference. In this work, we focus on the most generic form of numerical reasoning over text, encompassed by the reasoning-based MRC framework. A particularly challenging setting for this task is where the answers are numerical in nature as in the popular MRC dataset, DROP (Dua et al., 2019). Figure 1 shows the intricacies involved in the task, (*i*) passage and query language understanding, (*ii*) contextual understanding of the passage date and numbers, and (*iii*) application of quantitative reasoning (*e.g., max, not*) over dates and numbers to reach the final numerical answer.

Three broad genres of models have proven successful on the DROP numerical reasoning task. First, *large-scale pretrained language models* like GenBERT (Geva et al., 2020) uses a monolithic Transformer architecture and decodes numerical answers digit-by-digit. Though they deliver mediocre performance when trained only on the target data, their competency is derived from pretraining on massive synthetic data augmented with explicit supervision of the gold numerical reasoning. Second kind of models are the *reasoning-free hybrid models* like NumNet (Ran et al., 2019), NAQANet (Dua et al., 2019), NABERT+ (Kinley & Lin, 2019) and MTMSN (Hu et al., 2019), NeRd (Chen et al., 2020). They explicitly incorporate numerical computations in the standard extractive QA pipeline by learning a multi-type answer predictor over different reasoning types (*e.g., max/min, diff/sum, count, negate*) and directly predicting the corresponding numerical expression, instead of learning to reason. This is facilitated by exhaustively precomputing all possible outcomes of discrete operations and augmenting the training data with the reasoning-type supervision and numerical expressions that lead to the correct answer. Lastly, the most relevant class of models to consider for this work are the *modular networks for reasoning*. Neural Module Networks (NMN) (Gupta et al., 2020) is the first explicit reasoning based QA model which parses the query into a specialized program and executes it step-wise over learnable reasoning modules. However, to do so, apart from the exhaustive precomputation of all discrete operations, it also needs more fine-grained supervision of the gold

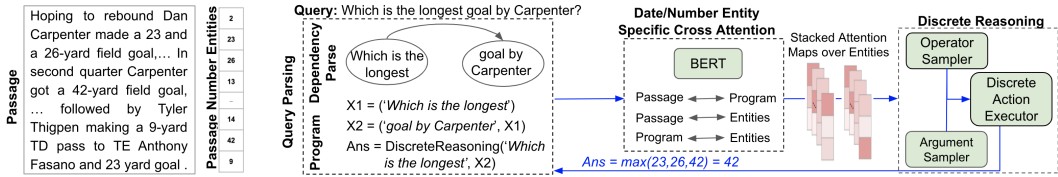

Figure 1: Example (passage, query, answer) from DROP and outline of our method: executing noisy program obtained from dependency parsing of query by learning date/number entity specific cross attention, and sampling and execution of discrete operations on entity arguments to reach the answer.

program and the gold program execution, obtained heuristically, by leveraging the abundance of templatized queries in DROP.

While being more pragmatic and richer at interpretability, both modular and hybrid networks are also tightly coupled with the additional supervision. For instance, the hybrid models cannot learn without it, and while NMN is the first to *enable* learning from QA pair alone, it still needs more finer-grained supervision for at least a part of the training data. With this, it manages to supercede the SoTA models NABERT and MTMSN on a carefully chosen subset of DROP using the supervision. However, NMN generalizes poorly to more open-ended settings where such supervision is not easy to handcraft.

**Need for symbolic reasoning.** One striking characteristic of the modular methods is to avoid discrete reasoning by employing only learnable modules with an exhaustively precomputed space of outputs. While they perform well on DROP, their modeling complexity grows arbitrarily with more complex non-linear numerical operations (*e.g.,* exp, log, cos). Contrarily, symbolic modular networks that execute the discrete operations are possibly more robust or pragmatic in this respect by remaining unaffected by the operation complexity. Such discrete reasoning has indeed been incorporated for simpler, well-structured tasks like math word problems (Koncel-Kedziorski et al., 2016) or KB/Table-QA (Zhong et al., 2017; Liang et al., 2018; Saha et al., 2019), with Deep Reinforcement Learning (RL) for end-to-end training. MRC however needs a more generalized framework of modular neural networks involving more fuzzy reasoning over noisy entities extracted from open-ended passages.

In view of this, we propose a **W**eakly-**S**upervised **N**euro-**S**ymbolic **M**odule **N**etwork (**WNSMN**)
• A first attempt at numerical reasoning based MRC, trained with answers as the sole supervision;
• Based on a generalized framework of dependency parsing of queries into noisy heuristic programs;
• End-to-end training of neuro-symbolic reasoning modules in a RL framework with discrete rewards;

To concretely compare WNSMN with contemporary NMN, consider the example in Figure 1. In comparison to our generalized query-parsing, NMN parses the query into a program form *(MAX(FILTER(FIND('Carpenter'), 'goal')))*, which is step-wise executed by different learnable modules with exhaustively precomputed output set. To train the network, it employs various forms of strong supervision such as gold program operations and gold query-span attention at each step of the program and gold execution *i.e.,* supervision of the passage numbers *(23, 26, 42)* to execute *MAX* operation on.

While NMN can only handle the 6 reasoning categories that the supervision was tailored to, WNSMN focuses on the full DROP with numerical answers (called DROP-*num*) that involves more diverse reasoning on more open-ended questions. We empirically compare WNSMN on DROP-*num* with the SoTA NMN and GenBERT that allow learning with partial or no strong supervision. Our results showcase that the proposed WNSMN achieves 32% better accuracy than NMN in absence of at least one or more types of supervision and performs 8% better than GenBERT when the latter is fine-tuned only on DROP in a comparable setup, without additional synthetic data having explicit supervision.

## 2 MODEL: WEAKLY SUPERVISED NEURO-SYMBOLIC MODULE NETWORK

We now describe our proposed WNSMN that learns to infer the answer based on weak supervision of the QA pair by generating the program form of the query and executing it through explicit reasoning.

**Parsing Query into Programs** To keep the framework generic, we use a simplified representation of the Stanford dependency parse tree (Chen & Manning, 2014) of the query to get a generalized program (Appendix A.5). First, a node is constructed for the subtree rooted at each child of the root by merging its descendants in the original word order. Next an edge is added from the left-most node (which we call the *root clause*) to every other node. Then by traversing left to right, each node is organized into a step of a program having a linear flow. For example, the program obtained in Figure

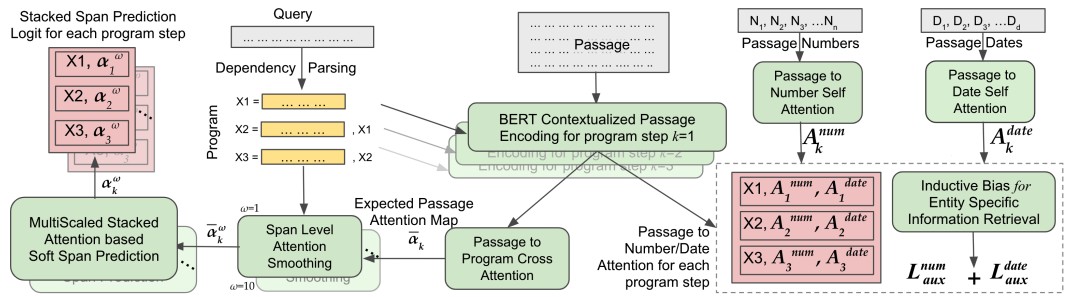

Figure 2: Modeling the interaction between the passage and (left) the program, & (right) its number/date entities. For each program step $k$, they respectively yield (i) Stacked Span Prediction Logits and (ii) Attention over Number/Date entities for each passage token. The linear combination of these two gives the expected distribution over entities, $\mathcal{T}_k^{num}$ and $\mathcal{T}_k^{date}$ for the step $k$

[1] is *X1 = ('which is the longest')*; *X2 = ('goal by Carpenter', X1)*; *Answer = Discrete-Reasoning('which is the longest', X2)*. Each program step consists of two types of arguments (*i*) Query Span Argument obtained from the corresponding node, indicates the query segment referred to, in that program step *e.g., 'goal by Carpenter'* in Step 2 (ii) Reference Argument(s) obtained from the incoming edges to that node, refers to the previous steps of the program that the current one depends on *e.g., X1* in Step 2. Next, a final step of the program is added, which has the reference argument as the leaf node(s) obtained in the above manner and the query span argument as the root-clause. This step is specifically responsible for handling the discrete operation, enabled by the root-clause which is often indicative of the kind of discrete reasoning involved (*e.g., max*). However this being a noisy heuristic, the QA model needs to be robust to such noise and additionally rely on the full query representation in order to predict the discrete operation. For simplicity we limit the number of reference arguments to 2.

## 2.1 PROGRAM EXECUTION

Our proposed WNSMN learns to execute the program over the passage in three steps. In the preprocessing step, it identifies numbers and dates from the passage, and maintains them as separate canonicalized entity-lists along with their mention locations. Next, it learns an entity-specific *cross-attention* model to rank the entities *w.r.t.* their query-relevance (§2.1.1), and then *samples* relevant entities as discrete arguments (§2.1.2) and executes appropriate discrete operations on them to reach the answer. An RL framework (§2.1.3) trains it end-to-end with the answer as the sole supervision.

### 2.1.1 ENTITY-SPECIFIC CROSS ATTENTION FOR INFORMATION EXTRACTION

To rank the query-relevant passage entities, we model the passage, program and entities jointly.

**Modeling interaction between program and passage** This module (Figure 2, left) learns to associate *query span* arguments of the program with the passage. For this, similar to NMN, we use a BERT-base pretrained encoder (Devlin et al., 2018) to get contextualized token embeddings of the passage and query span argument of each program step, respectively denoted by $\boldsymbol{P}_k$ and $\boldsymbol{Q}_k$ for the $k$'th program step. Based on it, we learn a *similarity* matrix $\mathbf{S} \in \mathbb{R}^{l \times n \times m}$ between the program and passage, where $l$, $n$, and $m$ respectively are the program length and query span argument and passage length (in tokens). Each $\boldsymbol{S}_k \in \mathbb{R}^{n \times m}$ represents the affinity over the passage tokens for the $k$'th program argument and is defined as $\boldsymbol{S}_k(i, j) = \boldsymbol{w}^T[\boldsymbol{Q}_{ki}; \boldsymbol{P}_{kj}; \boldsymbol{Q}_{ki} \odot \boldsymbol{P}_{kj}]$, where $\boldsymbol{w}$ is a learnable parameter and $\odot$ is element-wise multiplication. From this, an attention map $\boldsymbol{A}_k$ is computed over the passage tokens for the $k$'th program argument as $\boldsymbol{A}_k(i, j) = \text{softmax}_j(\boldsymbol{S}_k(i, j)) = \frac{\exp(\boldsymbol{S}_k(i,j))}{\sum_j \exp(\boldsymbol{S}_k(i,j))}$. Similarly, for the $i$'th token of the $k$'th program argument the cumulative attention $a_{ki}$ *w.r.t.* the passage is $a_{ki} = \text{softmax}_i(\sum_j \boldsymbol{S}_k(i, j))$. A linear combination of the attention map $\boldsymbol{A}_k(i, \cdot)$ weighted by $a_{ki}$ gives the expected passage attention for the $k$'th step, $\bar{\boldsymbol{\alpha}}_k = \sum_i a_{ki} \boldsymbol{A}_k(i, \cdot) \in \mathbb{R}^m$.

*Span-level smoothed attention.* To facilitate information spotting and extraction over contiguous spans of text, we regularize the passage attention so that the attention on a passage token is high if the attention over its neighbors is so. We achieve this by adopting a heuristic smoothing technique (Huang et al., 2020), taking a sliding window of different lengths $\omega = \{1, 2, \ldots 10\}$ over the passage,

and replacing the token-level attention with the attention averaged over the window. This results in 10 different attention maps over the passage for the $k$'th step of the program: $\{\bar{\boldsymbol{\alpha}}_k^\omega | \omega \in \{1, 2, \ldots, 10\}\}$.

*Soft span prediction.* This network takes a multi-scaled (Gupta et al., 2020) version of $\bar{\boldsymbol{\alpha}}_k^\omega$, by multiplying the attention map with $|\boldsymbol{s}|$ different scaling factors ($\boldsymbol{s} = \{1, 2, 5, 10\}$), yielding a $|\boldsymbol{s}|$-dimensional representation for each passage token, *i.e.,* $\bar{\boldsymbol{\alpha}}_k^\omega \in \mathbb{R}^{m \times |\boldsymbol{s}|}$. This is then passed through a $L$-layered stacked self-attention *transformer* block (Vaswani et al., 2017), which encodes it to $m \times d$ dimension, followed by a *linear layer* of dimension $d \times 1$, to obtain the span prediction logits: $\boldsymbol{\alpha}_k^\omega = Linear(Transformer(MultiScaling(\bar{\boldsymbol{\alpha}}_k^\omega))) \in \mathbb{R}^m$. Further the span prediction logits at each program step (say $k$) is additively combined with those from the previous steps referenced in the current one, through the reference argument ($ref(k)$) at step $k$, *i.e.,* $\boldsymbol{\alpha}_k^\omega = \boldsymbol{\alpha}_k^\omega + \sum_{k' \in ref(k)} \boldsymbol{\alpha}_{k'}^\omega$.

**Modeling interaction between program and number/date entities** This module (Figure 2, right) facilitates an entity-based information spotting capability, that is, given a passage mention of a number/date entity relevant to the query, the model should be able to attend to the neighborhood around it. To do this, for each program step, we first compute a *passage tokens to number tokens* attention map $\mathbf{A}^{num} \in \mathbb{R}^{l \times m \times N}$, where $N$ is the number of unique number entities. Note that this attention map is different for each program step as the contextual BERT encoding of the passage tokens ($\boldsymbol{P}_k$) is coupled with the program's span argument of that step. At the $k$-th step, the row $\boldsymbol{A}_k^{num}(i, \cdot)$ denotes the probability distribution over the $N$ unique number tokens *w.r.t.* the $i$-th passage token. The attention maps are obtained by a softmax normalization of each row of the corresponding *passage tokens to number tokens* similarity matrix, $\boldsymbol{S}_k^{num} \in \mathbb{R}^{m \times N}$ for $k = \{1 \ldots l\}$, where the elements of $\boldsymbol{S}_k^{num}$ are computed as $\boldsymbol{S}_k^{num}(i, j) = \boldsymbol{P}_{ki}^T \boldsymbol{W}_n \boldsymbol{P}_{kn_j}$ with $\boldsymbol{W}_n \in \mathbb{R}^{d \times d}$ being a learnable projection matrix and $n_j$ being the passage location of the $j$-th number token. These similarity scores are additively aggregated over all mentions of the same number entity in the passage.

The relation between program and entities is then modeled as $\boldsymbol{\tau}_k^\omega = \text{softmax}(\sum_i \alpha_{ki}^\omega \boldsymbol{A}_k^{num}(i, \cdot)) \in \mathbb{R}^N$, which gives the expected distribution over the $N$ number tokens for the $k$-th program step and using $\omega$ as the smoothing window size. The final stacked attention map obtained for the different windows is $\mathcal{T}_k^{num} = \{\boldsymbol{\tau}_k^\omega | \omega \in \{1, 2, \ldots 10\}\}$. Similarly, for each program step $k$, we also compute a separate stacked attention map $\mathcal{T}_k^{date}$ over the unique date tokens, parameterized by a different $\boldsymbol{W}_d$.

A critical requirement for a meaningful attention over entities is to incorporate information extraction capability in the number and date attention maps $\mathbf{A}^{num}$ and $\mathbf{A}^{date}$, by enabling the model to attend over the neighborhood of the relevant entity mentions. This is achieved by minimizing the unsupervised auxiliary losses $\mathcal{L}_{aux}^{num}$ and $\mathcal{L}_{aux}^{date}$ in the training objective, which impose an inductive bias over the number and date entities, similar to Gupta et al. (2020). Its purpose is to ensure that the passage attention is densely distributed inside the neighborhood of $\pm \Omega$ (a hyperparameter, *e.g.,* 10) of the passage location of the entity mention, without imposing any bias on the attention distribution outside the neighborhood. Consequently, it maximises the log-form of cumulative likelihood of the attention distribution inside the window and the entropy of the attention distribution outside of it.

$$\mathcal{L}_{aux}^{num} = -\frac{1}{l} \sum_{k=1}^l \left[ \sum_{i=1}^m \left[ \log(\sum_{j=1}^N \mathbb{1}_{n_j \in [i \pm \Omega]} a_{kij}^{num}) - \sum_{j=1}^N \mathbb{1}_{n_j \notin [i \pm \Omega]} a_{kij}^{num} \log(a_{kij}^{num}) \right] \right] \quad (1)$$

where $\mathbb{1}$ is indicator function and $a_{kij}^{num} = \boldsymbol{A}_k^{num}(i, j)$. $\mathcal{L}_{aux}^{date}$ for date entities is similarly defined.

### 2.1.2 MODELING DISCRETE REASONING

The model next learns to execute a single step[1] of discrete reasoning (Figure 3) based on the final program step. The final step contains (*i*) root-clause of the query which often indicates the type of discrete operation (*e.g., 'what is the longest'* indicates max, *'how many goals'* indicates count), and (*ii*) reference argument indicating the previous program steps the final step depends on. Each previous step (say $k$) is represented as stacked attention maps $\mathcal{T}_k^{num}$ and $\mathcal{T}_k^{date}$, obtained from §2.1.1.

**Operator Sampling Network** Owing to the noisy nature of the program, the operator network takes as input: (*i*) BERT's [CLS] representation for the passage-query pair and LSTM (Hochreiter & Schmidhuber, 1997) encoding (randomly initialized) of the BERT contextual representation of (*ii*) the root-clause from the final program step and (*iii*) full query (*w.r.t.* passage), to make two predictions:

---

[1]This is a reasonable assumption for DROP with a recall of 90% on the training set. However, it does not limit the generalizability of WNSMN, as with standard beam search it is possible to scale to an $l$-step MDP.

Figure 3: Operator & Argument Sampling Network and RL framework over sampled discrete actions

- *Entity-Type Predictor Network*, an Exponential Linear Unit (Elu) activated fully-connected layer followed by a $\mathrm{softmax}$ that outputs the probabilities of sampling either date or number types.
- *Operator Predictor Network*, a similar Elu-activated fully connected layer followed by a $\mathrm{softmax}$ which learns a probability distribution over a fixed catalog of 6 numerical and logical operations ($\mathrm{count, max, min, sum, diff, negate}$), each represented with learnable embeddings.

Apart from the $\mathrm{diff}$ operator which acts only on two arguments, all other operations can take any arbitrary number of arguments. Also some of these operations can be applied only on numbers (*e.g.,* $\mathrm{sum, negate}$) while others can be applied on both numbers or date (*e.g.,* $\mathrm{max, count}$).

**Argument Sampling Network** This network learns to sample date/number entities as arguments for the sampled discrete operation, given the entity-specific stacked attentions ($\mathcal{T}_k^{num}$ and $\mathcal{T}_k^{date}$) for each previous step (say, $k$), that appears in the reference argument of the final program step. In order to allow sampling of fixed or arbitrary number of arguments, the argument sampler learns four types of networks, each modeled with a $L$-layered stacked self attention based $Transformer$ block (with output dimension $d$) followed by different non-linear layers embodying their functionality and a $\mathrm{softmax}$ normalization to get the corresponding probability of the argument sampling (Figure 3).

- Sample $n \in \{1, 2\}$ Argument Module: $\mathrm{softmax}(Elu(Linear_{d \times n}(Transformer(\mathcal{T}))))$, outputs a distribution over the single entities ($n = 1$) or a joint distribution over the entity-pairs ($n = 2$).
- Counter Module: $\mathrm{softmax}(Elu(Linear_{d \times 10}(CNN\text{-}Encoder(Transformer(\mathcal{T})))))$, predicts a distribution over possible count values ($\in [1, \dots, 10]$) of number of entity arguments to sample.
- Entity-Ranker Module: $\mathrm{softmax}(PRelu(Linear_{d \times 1}(Transformer(\mathcal{T}))))$, learns to rerank the entities and outputs a distribution over all the entities given the stacked attention maps as input.
- Sample Arbitrary Argument: $Multinomial$(Entity-Ranked Distribution, Counter Prediction).

Depending on the number of arguments needed by the discrete operation and the number of reference arguments in the final program step, the model invokes one of *Sample {1, 2, Arbitrary} Argument*. For instance, if the sampled operator is $\mathrm{diff}$ which needs 2 arguments, and the final step has 1 or 2 reference arguments, then the model respectively invokes either *Sample 2 argument* or *Sample 1 argument* on the stacked attention $\mathcal{T}$ corresponding to each reference argument. And, for operations needing arbitrary number of arguments, the model invokes the *Sampling Arbitrary Argument*. For the *Arbitrary Argument* case, the model first predicts the number of entities $c \in \{1, \dots, 10\}$ to sample using the Counter Network, and then samples from the multinomial distribution based on the joint of $c$-combinations of entities constructed from the output distribution of the Entity Ranker module.

### 2.1.3 Training with Weak Supervision in the Deep RL Framework

We use an RL framework to train the model with only discrete binary feedback from the exact match of the gold and predicted numerical answer. In particular, we use the REINFORCE (Williams, 1992) policy gradient method where a stochastic policy comprising a sequence of actions is learned with the goal of maximizing the expected reward. In our case, the discrete operation along with argument sampling constitute the *action*. However, because of our assumption that a single step of discrete reasoning suffices for most questions in DROP, we further simplify the RL framework to a contextual multi-arm bandit (MAB) problem with a 1-step MDP, *i.e.,* the agent performs only one step action.

Despite the simplifying assumption of 1-step MDP, the following characteristics of the problem render it highly challenging: (*i*) the action space $\mathcal{A}$ is exponential in the order of number of operations and argument entities in the passage (averaging to *12K* actions for DROP-*num*); (*ii*) the extreme reward sparsity owing to the binary feedback is further exacerbated by the presence of spurious rewards, as the same answer can be generated by multiple diverse actions. Note that previous approaches like

NMN can avoid such spurious supervision because they heuristically obtain additional annotation of the question category, the gold program or gold program execution atleast for some training instances.

In our contextual MAB framework, for an input $x = (\text{passage}(p), \text{query}(q))$, the context or environment state $s_\phi(x)$ is modeled by the entity specific cross attention (§2.1.1, parameterized by $\phi$) between the (*i*) passage (*ii*) program-form of the query and (*iii*) extracted passage date/number entities. Given the state $s_\phi(x)$, the layout policy (§2.1.2, parameterized by $\theta$) then learns the query-specific inference layout, *i.e.,* the discrete action sampling policy $P_\theta(a|s_\phi(x))$ for action $a \in \mathcal{A}$. The action sampling probability is a product of the probability of sampling entities from the appropriate entity type ($P_\theta^{type}$), probability of sampling the operator ($P_\theta^{op}$), and probability of sampling the entity argument(s) ($P_\theta^{arg}$), normalized by number of arguments to sample. Therefore, with the learnable context representation $s_\phi(x)$ of input $x$, the end-to-end objective is to jointly learn $\{\theta, \phi\}$ that maximises the expected reward $R(x, a) \in \{-1, +1\}$ over the sampled actions ($a$), based on exact match with the gold answer.

To mitigate the learning instability in such sparse confounding reward settings, we intialize with a simpler iterative *hard-Expectation Maximization (EM)* learning objective, called Iterative Maximal Likelihood (IML) (Liang et al., 2017). With the assumption that the sampled actions are extensive enough to contain the gold answer, IML greedily searches for the *good* actions by fixing the policy parameters, and then maximises the likelihood of the *best* action that led to the highest reward. We define *good* actions ($\mathcal{A}^{good}$) as those that result in the gold answer itself and take a conservative approach of defining *best* among them as simply the most likely one according to the current policy.

$$J^{IML}(\theta, \phi) = \sum_x \max_{a \in \mathcal{A}^{good}} \log P_{\theta, \phi}(a|x) \tag{2}$$

After the IML initialization, we switch to REINFORCE as the learning objective after a few epochs, where the goal is to maximise the expected reward ($J^{RL}(\theta, \phi) = \sum_x \mathbb{E}_{P_{\theta, \phi}(a|x)} R(x, a)$) as

$$\nabla_{(\theta, \phi)} J^{RL} = \sum_x \sum_{a \in \mathcal{A}} P_{\theta, \phi}(a|x)(R(x, a) - B(x))\nabla_{\theta, \phi}(\log P_{\theta, \phi}(a|x)) \tag{3}$$

where $B(x)$ is simply the average (baseline) reward obtained by the policy for that instance $x$. Further, in order to mitigate overfitting, in addition to $L_2$-regularization and dropout, we also add entropy based regularization over the argument sampling distribution, in each of the sampling networks.

## 3 EXPERIMENTS

We now empirically compare the *exact-match* performance of WNSMN with SoTA baselines on versions of DROP dataset and also examine how it fares in comparison to strong supervised skylines. The **Primary Baselines** for WNSMN are the explicit reasoning based **NMN** (Gupta et al., 2020) which uses additional strong supervision and the BERT based language model **GenBERT** (Geva et al., 2020) that does not embody any reasoning and autoregressively generates numeric answer tokens. As the **Primary Dataset** we use **DROP-*num***, the subset of DROP with numerical answers. This subset contains 45K and 5.8K instances respectively from the standard DROP train and development sets. Originally, NMN was showcased on a very specific subset of DROP, restricted to the 6 reasoning-types it could handle, out of which three (*count*, *date-difference*, *extract-number*) have numeric answers. This subset comprises 20K training and 1.8K development instances, out of which only 10K and 800 instances respectively have numerical answers. We further evaluate on this numerical subset, referred to as **DROP-Pruned-*num***. In both the cases, the training data was randomly split into 70%:30% for train and internal validation and the standard DROP development set was treated as the Test set.

Figure 4 shows the t-SNE plot of pretrained Sentence-BERT (Reimers & Gurevych, 2019) encoding of *all* questions in DROP-*num*-Test and also the DROP-Pruned-*num*-Test subset with different colors (red, green, yellow) representing different types. Not only are the DROP-*num* questions more diverse than the carefully chosen DROP-Pruned-*num* subset, the latter also forms well-separated clusters corresponding to the three reasoning types. Additionally, the average perplexity (using `nltk`) of the DROP-Pruned-*num* and DROP-*num* questions was found to be *3.9* and *10.65* respectively, further indicating the comparatively open-ended nature of the former.

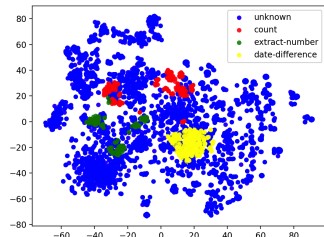

Figure 4: t-SNE plot of DROP-*num*-Test questions.

For the primary baselines NMN and GenBERT, we report the performance on in-house trained models on the respective datasets, using

the code open-sourced by the authors. The remaining results, taken from Geva et al. (2020), Kinley & Lin (2019), and Ran et al. (2019); refer to models trained on the full DROP dataset. All models use the same pretrained BERT-base. Also note that a primary requirement of all models other than GenBERT and WNSMN *i.e.,* for NMN, MTMSN, NABERT, NAQANET, NumNet, is the exhaustive enumeration of the output space of all possible discrete operations. This simplifies the QA task to a classification setting, thus alleviating the need for discrete reasoning in the inference processs.

Table 1 presents our primary results on **DROP-*num***, comparing the performance of WNSMN (accuracy of the top-1 sampled action by the RL agent) with various ablations of NMN (provided in the authors' implementation) by removing atleast one of **Prog**ram, **Exec**ution, and **Q**uery **Att**ention supervision (Appendix A.4.1) and GenBERT models with pretrained BERT that are finetuned on DROP or DROP-*num* (denoted as GenBERT and GenBERT-*num*). For a fair comparison with our weakly supervised model, we do not treat NMN with all forms of supervision or GenBERT model pretrained with additional *synthetic* numerical and textual data as comparable baselines. Note that these GenBERT variants indeed enjoy strong reasoning supervision in terms of gold arithmetic expressions provided in these auxiliary datasets.

Table 1: DROP-*num*-Test Performance of Baselines and WNSMN.

| Supervision Type | | | Acc. (%) |
|---|---|---|---|
| Prog. | Exec. | QAtt. | |
| **NMN-*num* variants** | | | |
| ✗ | ✓ | ✓ | 11.77 |
| ✓ | ✗ | ✓ | 17.52 |
| ✓ | ✓ | ✗ | 18.27 |
| ✓ | ✗ | ✗ | 18.54 |
| ✗ | ✓ | ✗ | 12.27 |
| ✗ | ✗ | ✓ | 11.80 |
| ✗ | ✗ | ✗ | 11.70 |
| **GenBERT** | | | |
| ✗ | ✗ | ✗ | 42.30 |
| **GenBERT-*num*** | | | |
| ✗ | ✗ | ✗ | 38.41 |
| **WNSMN** | | | |
| ✗ | ✗ | ✗ | **50.97** |

NMN's performance is abysmally poor, indeed a drastic degradation in comparison to its performance on the pruned DROP subset reported by Gupta et al. (2020) and in our subsequent experiments in Table 2. This can be attributed to their limitation in handling more diverse classes of reasoning and open-ended queries in DROP-*num*, further exacerbated by the lack of one or more types of strong supervision.[2] Our earlier analysis on the complexity of the questions in the subset and full DROP-*num* further quantify the relative difficulty level of the latter. On the other hand, GenBERT delivers a mediocre performance, while GenBERT-*num* degrades additionally by 4%, as learning from numerical answers alone further curbs the language modeling ability. Our model performs significantly better than both these baselines, surpassing GenBERT by 8% and the NMN baseline by around 32%. This showcases the significance of incorporating explicit reasoning in neural models in comparison to the vanila large scale LMs like GenBERT. It also establishes the generalizability of such reasoning based models to more open-ended forms of QA, in comparison to contemporary modular networks like NMN, owing to its ability to handle both learnable and discrete modules in an end-to-end manner.

Next, in Table 2, we compare the performance of the proposed WNSMN with the same NMN variants (as in Table 1) on **DROP-Pruned-*num***. Some of the salient observations are: (*i*) WNSMN in fact reaches a performance quite close to the *strongly supervised* NMN variant (first row), and is able to attain at least an improvement margin of 4% over all other variants obtained by removing one or more types of supervision. This is despite all variants of NMN *additionally* enjoying the exhaustive precomputation of the output space of possible numerical answers; (*ii*) WNSMN suffers only in the case of *extract-number* type operations (*e.g., max,min*) that involve a more complex process of sampling arbitrary number of arguments (*iii*) Performance drop of NMN is not very large when all or none of the strong supervision is present, possi-

Table 2: DROP-Pruned-*num*-Test Performance of NMN variants and WNSMN

| Supervision-Type | | | Acc. (%) | Count | Extract-num | Date-differ |
|---|---|---|---|---|---|---|
| Prog. | Exec. | QAtt. | | | | |
| **NMN-*num* Variants** | | | | | | |
| ✓ | ✓ | ✓ | **68.6** | 50.0 | **88.4** | 72.5 |
| ✗ | ✓ | ✓ | 42.4 | 24.1 | 73.9 | 36.4 |
| ✓ | ✗ | ✓ | 54.3 | 47.9 | 80.7 | 40.9 |
| ✓ | ✓ | ✗ | 63.3 | 45.5 | 81.1 | 68.7 |
| ✗ | ✗ | ✓ | 48.2 | 38.1 | 72.4 | 41.9 |
| ✓ | ✗ | ✗ | 61.0 | 44.7 | 81.1 | 63.2 |
| ✗ | ✓ | ✗ | 62.3 | 43.7 | 84.1 | 67.7 |
| ✗ | ✗ | ✗ | 62.1 | 46.8 | 83.6 | 66.1 |
| **WNSMN** | | | | | | |
| ✗ | ✗ | ✗ | 66.5 | **58.8** | 66.8 | **75.2** |

bly because of the limited diversity over reasoning types and query language; and (*iv*) Query-Attention supervision infact adversely affects NMN's performance, in absence of the *program* and *execution* supervision or both, possibly owing to an undesirable biasing effect. However when both supervisions are available, query-attention is able to improve the model performance by 5%. Further, we believe the test set of 800 instances is too small to get an unbiased reflection of the model's performances.

---

[2]Both the results and limitations of NMN in Table 1 and 2 were confirmed by the authors of NMN as well.

In Table 3, we additionally inspect recall over the top-$k$ actions sampled by WNSMN to estimate how it fares in comparison to the strongly supervised skylines: (*i*) NMN with all forms of strong supervision; (*ii*) GenBERT variants +ND, +TD and +ND+TD further pretrained on synthetic **N**umerical and **T**extual **D**ata and both; (*iii*) reasoning-free hybrid models like MTMSN (Hu et al., 2019) and NumNet (Ran et al., 2019), NAQANet (Dua et al., 2019) and NABERT, NABERT+ (Kinley & Lin, 2019). Note that both NumNet and NAQANet do not use pretrained BERT. MTMSN achieves SoTA performance through a supervised framework of training specialized predictors for each reasoning type to predict the numerical expression directly instead of learning to reason. While top-1 performance of WNSMN (in Table 1) is $4\%$ worser than NABERT, Recall@top-2 is equivalent to the strongly supervised NMN, top-5 and top-10 is comparable to NABERT+, NumNet and GenBERT models +ND, +TD and top-20 nearly achieves SoTA. Such promising recall over the top-$k$ actions suggests that more sophisticated RL algorithms with better exploration strategies can possibly bridge this performance gap.

## 4 ANALYSIS & FUTURE WORK

**Performance Analysis** Despite the notorious instabilities of RL due to high variance, the training trend, as shown in Figure 5(a) is not afflicted by catastrophic forgetting. The sudden performance jump between epochs 10-15 is because of switching from iterative ML initialization to REINFORCE objective. Figure 5(b) shows the individual module-wise performance evaluated using the noisy pseudo-rewards, that indicate whether the action sampled by this module *led* to the correct answer or not (details in Appendix A.6). Further, by bucketing the performance by the total number of passage entities in Figure 5(c), we observe that WNSMN remains unimpacted by the increasing number of date/numbers, despite the action space explosion. On the other hand, GenBERT's performance drops linearly beyond 25 passage entities and NMN-*num* degrades exponentially from the beginning, owing to its direct dependency on the exponentially growing exhaustively precomputed output space.

Table 3: Skylines & WNSMN top-$k$ performance on DROP-*num*-Test

| Strongly Supervised Models | Acc. (%) |
|---|---|
| NMN-*num* (all supervision) | 58.10 |
| GenBERT+ND | 69.20 |
| GenBERT+TD | 70.50 |
| GenBERT+ND+TD | 75.20 |
| NAQANet | 44.97 |
| NABERT | 54.27 |
| NABERT+ | 66.60 |
| NumNet | 69.74 |
| MTMSN | 75.00 |

| Recall@top-$k$ actions of WNSMN (%) | | | | | |
|---|---|---|---|---|---|
| $k=2$ | $k=3$ | $k=4$ | $k=5$ | $k=10$ | $k=20$ |
| 58.6 | 63.0 | 65.4 | 67.4 | 72.3 | 74.2 |

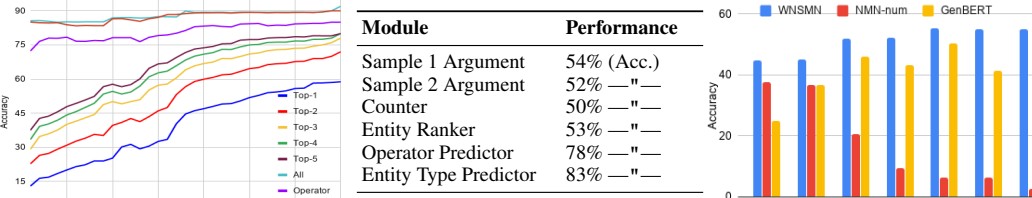

Figure 5: (a) Training trend showing the Recall@top-$k$ and all actions, accuracy of Operator and Entity-type Predictor, estimated based on noisy psuedo rewards (Appendix A.6), (b) Module-wise performance (using pseudo-reward) on DROP-*num*-Test, (c) Bucketing performance by total number of passage entities for WNSMN, and the best performing NMN and GenBERT model from Table 1.

**More Stable RL Framework** The training trend in Figure 5(a) shows early saturation and the module-wise performance indicates overfitting despite the regularization tricks in §2.1.3 and Appendix A.6. While more stable RL algorithms like Actor-Critic, Trust Region Policy Optimization (Schulman et al., 2015) or Memory Augmented Policy Optimization (Liang et al., 2018) can mitigate these issues, we leave them for future exploration. Also, though this work's objective was to train module networks with weak supervision, the sparse confounding rewards in the exponential action space indeed render the RL training quite challenging. One practical future direction to bridge the performance gap would be to pretrain with strong supervision on at least a subset of reasoning categories or on more constrained forms of synthetic questions, similar to GenBERT. Such a setting would require inspection and evaluation of generalizability of the RL model to unknown reasoning types or more open-ended questions.

## 5 RELATED WORK

In this section we briefly compare our proposed WNSMN to the two closest genre of models that have proven quite successful on DROP [3] i) reasoning free hybrid models NumNet, NAQANet, NABERT, NABERT+, MTMSN, and NeRd ii) modular network for reasoning NMN. Their main distinction with WNSMN is that in order to address the challenges of weak supervision, they obtain program annotation from the QA pairs through i) various heuristic parsing of the templatized queries in DROP to get supervision of the reasoning type (max/min, diff/sum, count, negate). ii) exhaustive search over all possible discrete operations to get supervision of the arguments in the reasoning.

Such heuristic supervision makes the learning problem significantly simpler in the following ways

- These models enjoy supervision of specialized program that have explicit information of the type of reasoning to apply for a question *e.g.,* SUM(10,12)
- A simplistic (contextual BERT-like) *reader* model to read query related information from the passage trained with direct supervision of the query span arguments at each step of the program
- A *programmer* model that can be directly trained to decode the specialized programs
- *Executing* numerical functions (*e.g., difference, count, max, min*) either by i) training purely neural modules in a strong supervised setting using the annotated programs or by ii) performing the actual discrete operation as a post processing step on the model's predicted program. For each of these previous works, it is possible to directly apply the learning objective on the space of decoded program, without having to deal with the discrete answer or any non-differentiability.

However, such heuristic techniques of program annotation or exhaustive search is not practical as the language of questions or the space of discrete operations become more complex. Hence WNSMN learns in the challenging weak-supervised setting without any additional annotation through

- A noisy symbolic query decomposition that is oblivious to the reasoning type and simply based on generic text parsing techniques
- An entity specific cross attention model extracting passage information relevant to each step of the decomposed query and learning an attention distribution over the entities of each type
- Learning to apply discrete reasoning by employing neural modules that learn to sample the operation and the entity arguments
- Leveraging a combination of neural and discrete modules when executing the discrete operation, instead of using only neural modules which need strong supervision of the programs for learning the functionality
- Fundamentally different learning strategy by incorporating inductive bias through auxiliary losses and Iterative Maximal Likelihood for a more conservative initialization followed by REINFORCE

These reasoning-free hybrid models are not comparable with WNSMN because of their inability to learn in absence of any heuristic program annotation. Instead of learning to reason based on only the final answer supervision, they reduce the task to learning to decode the program, based on heuristic program annotation. NMN is the only reasoning based model that employ various auxiliary losses to learn even in absence of any additional supervision, similar to us.

To our knowledge WNSMN is the first work on modular networks for fuzzy reasoning over text in RC framework, to handle the challenging cold start problem of the weak supervised setting without needing any additional specialized supervision of heuristic programs.

## 6 CONCLUSION

In this work, we presented Weakly Supervised Neuro-Symbolic Module Network  for numerical reasoning based MRC based on a generalized framework of query parsing to noisy heuristic programs. It trains both neural and discrete reasoning modules end-to-end in a Deep RL framework with only discrete reward based on exact answer match. Our empirical analysis on the *numerical-answer only subset* of DROP showcases significant performance improvement of the proposed model over SoTA NMNs and Transformer based language model GenBERT, when trained in comparable weakly supervised settings. While, to our knowledge, this is the first effort towards training modular networks for fuzzy reasoning over RC in a weakly-supervised setting, there is significant scope of improvement, such as employing more sophisticated RL framework or by leveraging the pretraining of reasoning.

---

[3]A more detailed related work section is presented in the Appendix A.4

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
