# OpenReview forum: "Weakly Supervised Neuro-Symbolic Module Networks for Numerical Reasoning"
_ICLR.cc/2021/Conference — Reject_

### Official Review · AnonReviewer1 · 2020-10-28
**A well written paper with good experiment results.**

**Rating:** 6
**Confidence:** 2

**Review:**

This paper proposed a weakly supervised module networks to solve numeric reasoning problems. The model is trained with reinforcement learning over the modules. The model solved a few challenges in training the RL model. They evaluated their model on a subset of the DROP dataset.

This paper was clearly presented and the results was promising. My only concern is that they only evaluate the model on a carefully constructed subset. I am not sure how it compare to other models in more general settings and/or on other datasets.

The authors mentioned that most questions in the dataset only requires 1-hop reasoning, so they reduced their model to 1-step MDP. Will your model work on questions that involves multi-step reasoning? Can you show that?

Did you detect the query span as a preprocessing step? What if the query spans are noisy/missing/duplicated?

* I did not check the correctness of the equations.

---

> ### Author Response · Authors · 2020-11-18
> **Addressing questions on generalizability**
>
> We thank the reviewer for the feedback and comments and try to address the questions raised as follows:
>
> ----------------------------------------------------------------------------------------------------------------------------------------------------
>
> Question: WNSMN model is evaluated on a carefully constructed subset?
>
> Response: Since the focus of our work is numerical reasoning, we train and evaluate the model on the subset of DROP with numerical answers. No other considerations were made when deciding the subset.
>
> ----------------------------------------------------------------------------------------------------------------------------------------------------
>
> Question: Reason for focusing on numerical problems?
>
> Response: Since in this work we are investigating how neural models can incorporate explicit reasoning especially in the challenging weak supervised setting, we focus on only answering questions having numerical answer. We believe the effect of explicit reasoning is more directly observable for such questions.
>
> This is backed up by the category-wise performance comparison of reasoning-free language model GenBERT (reported in Geva et al. (2020)) with other hybrid models (MTMSN and NABERT+) that exploit numerical computation required in answering DROP questions. While, onDROP-num, there is an accuracy gap of 33% between the GenBERT model and the hybrid models(when all are trained on DROP only), there is only a 2-3% performance gap on the subset having answers as single span, despite the latter also needing reasoning. This evinces that the performance gap is indeed due to exploiting explicit reasoning under such strong supervised settings.
>
> ----------------------------------------------------------------------------------------------------------------------------------------------------
>
> Question: Concerns with generalizability
>
> Response: In order to work on other kinds of named entities like PERSON or NOUN PHRASE, only the first preprocessing step of extracting relevant entities from the passage would differ. Instead of identifying numbers and dates, it would need a way to symbolically define entities (can be numbers, dates or named entities like PERSON) and link their mentions in the passage to canonicalized entity names.
>
> Once separate entity-lists along with their mention locations have been created for each required entity-type, the same Entity-Specific Cross Attention for Information Extraction and Discrete Reasoning steps described in Sec 2.1 can be followed to reach the answer.
>
> As long as the desired discrete operations can be defined on other named entity types and have an output space comparable to the final answer (e.g.,space of numerical quantities) which allows us to define some reward function, the same architecture can be used on other entity types.
>
>
> ----------------------------------------------------------------------------------------------------------------------------------------------------
>
> Question: Will your model work on questions that involve multi-step reasoning?
>
> Response: For the current work, we assume that the last step of the query decomposition needs discrete reasoning (though, through the program structure and the entity-specific cross attention over each step, the last step accumulates attention over the previous steps).
> For multi-step reasoning
> - The discrete reasoning module can be invoked for each step of the query decomposition
> - Every step of the discrete reasoning can output a probability distribution over possible discrete actions taken at that step
> - The reference argument at each step of discrete reasoning which refers to one or more previous steps, can take the output of the (top-1 or top-K in beam search setting) action taken in the referred step and accordingly incorporate the action probabilities multiplicatively.
> - Reward w.r.t the gold answer is computed based on the top-K actions at the final step only.
>
> This needs handling of sequence of actions (as opposed to a contextual Bandit). Though our framework can generalize to that, the action space expansion can lead to more learning challenges.This can certainly be investigated in future and may require pre-training of the reasoning (using data augmentation like in GenBERT) to keep the learning tractable. We are also not aware of any dataset for numerical reasoning over text which predominately requires multi-step reasoning in the RC framework.

---

> > ### Author Response · Authors · 2020-11-18
> > **Addressing remaining questions**
> >
> > Question: Did you detect the query span as a preprocessing step? What if the query spans are noisy/missing/duplicated?
> >
> > Response: The query span arguments are obtained from the noisy query decomposition using generic parsing techniques as a preprocessing step. However one of the motivations of this work itself is to be able to learn the nature of discrete reasoning by predicting the discrete operation and the discrete arguments from this noisy query decomposition. Note that these discrete sampling networks take as input not only the noisy program structure but also the original query. The model can implicitly learn whether(or to what extent) to depend on the full query or the query decomposition for different instances. Similarly the argument sampling networks learn how to manipulate the attention over entities for each step of the noisy query decomposition, to sample the correct entities for the discrete reasoning.
> >
> > In order to make learning tractable in our weak supervised setting, we take this approach of combining both neural and symbolic tools in processing the query, but make provisions in the former to also directly learn from the original input since the latter can be noisy. Our experiments on the DROP-num dataset show this process is quite robust.
> >
> > ------------------------------------------------------------------------------------------------------------------------------------------
> >
> >
> > Question: Qualitative Analysis of WNSMN predictions
> >
> > Response: A manual analysis of WNSMN predictions of 334 instances of DROP-num test questions where the gold answer is obtained by multiple (>1) actions sampled by WNSMN (including the top-1 action)has been added in Appendix section A.3 (section on ‘Qualitative Inspection of WNSMN predictions’).Only 28 instances were found to have spurious action in top-1.

---

### Official Review · AnonReviewer3 · 2020-10-28
**Missing critical comparison to NeRd**

**Rating:** 4
**Confidence:** 4

**Review:**

This paper proposes a neurosymbolic module network that predicts a program structure following a dependency parse, populates that program's arguments, and executes it to answer numerical reasoning questions over text.  They claim that compared to Gupta et al. (2020), this approach doesn't require as many domain-specific heuristics to find gold programs or as much precomputation -- it is learned with weak supervision only (just the answers). The model has a number of pieces allowing the model to reference entities, numbers, and dates in a cross-attentive fashion. Results show that on numerical questions from the DROP dataset, the model outperforms that of Gupta et al. and is competitive with other approaches when appropriate assumptions are made.

In general, I like what this paper is trying to do. In a vacuum, I might recommend accepting it. However, my most pressing concern with this paper is the fairly fundamental similarity to this ICLR 2020 spotlight paper:

https://openreview.net/pdf?id=ryxjnREFwH
Neural Symbolic Reader: Scalable Integration of Distributed and Symbolic Representations for Reading Comprehension
Xinyun Chen, Chen Liang, Adams Wei Yu, Denny Zhou, Dawn Song, Quoc V. Le

I will refer to this paper as Chen et al. and their system as NeRd.

The authors failed to cite this paper, so I presume they are not aware of it. There are many similarities, including the broad strokes behind the DSL and the learning approach with weak supervision. I will discuss this current paper in contrast to that one to highlight the differences and the contributions.

Model: My favorite aspect of the current work is the fact that it leverages dependency representations. I like this idea a lot and buy that it might be a path to more general models of this form. However, it's hard for me to evaluate whether this is really more general due to the evaluation focused on this dataset, so this contribution feels somewhat theoretical. Would this dependency-based formalism work well elsewhere, or does it just happen to work well for DROP? It feels hard to make this work for larger, more compositional problems where the syntax-semantics divide is going to manifest itself more strongly (e.g., surrounding quantification).

The thing I like least about this paper is the complexity of the model.  The NeRd programmer is substantially simpler, as a basic seq2seq model. While you can argue that they have a more aggressive approach for warm-starting, like Gupta et al., there's less inductive bias built into the model directly.  By contrast, the current paper has a large number of bespoke modules for handling things from the DROP dataset.  I'm okay with things like treating entities as first-class concepts and allowing attention over them. But specialized attention maps targeting numbers and dates starts to feel very tailored to this dataset. Given the focus on numerical reasoning, perhaps some of this is to be expected, but I believe the other past approaches (NeRd, Gupta) have architectures that are a bit more general.

DSL: The DSLs between these two approaches are quite similar, but this is to be expected as the task is the same.

Weak supervision: Chen et al. use some heuristics as well as data augmentation to start learning off by giving the model access to a larger number of programs which find the right answer right off the bat. To handle spurious programs, they use hard EM wth an additional thresholding step to avoid training on programs that are extremely unlikely.  The present work initializes with a hard EM-style learning before switching to REINFORCE. In my view, there is not much of a conceptual advantage to this approach over NeRd. I don't think this paper has a particularly keen insight to solve the cold-start problem for RL here, and the NeRd-style data augmentation is not that big of a weakness. So I don't see a fundamental contribution here over NeRd.

Results: Table 1 looks convincing but I'm having a hard time fully understanding it. It seems like the low numbers for NMN in DROP-num-Test are because this includes many question types explicitly excluded by the original Gupta et al. paper -- is this correct? This is fair but I think should be more explicit in the table.

As for Table 2, the ablations of different attention methods are interesting, but I'm not sure this is a fully fair comparison to Gupta et al. These pieces can't be mapped in an apples-to-apples fashion to aspects of the current approach. Gupta et al.  could just as easily say: well, let me ablate your specialized date handling module and do a comparison on that axis. In a new setting, we can claim such sources of supervision might be unavailable, but it's not clear to me that the system with this component deleted is the fairest point of comparison.

It also seems based on the Table 3 comparisons, we do not expect WNSMN to perform well compared to NeRd (see Table 4 in that paper).

OVERALL

Normally I am not a stickler for novelty over prior work. However, in this case, I feel compelled to judge this paper and directly compare it to Chen et al., as the motivation, evaluation conditions, and technical details of these papers bear striking similarities. (That paper also evaluates on MathQA, which is another point in its favor.) And in summary, this work does not offer enough new material to justify acceptance.  It is quite well-done work and has some interesting insights, though it suffers from an overly complex neural model. But the insights are not new, and the results are not enough to convince me that it should be accepted.

---

> ### Author Response · Authors · 2020-11-18
> **Comparison with NeRd model**
>
> We thank the Reviewer for the detailed feedback and observations. In the following, we address the questions/comments raised.
>
> -----------------------------------------------------------
>
> Question: Comparison with NeRd
>
> Response: First, we thank the reviewer for pointing out the relevant related work of NeRd which we had missed. However, after careful consideration of the paper, we humbly disagree with our reviewer on some key arguments as we explain below.
>
> The main distinction of NeRd with our work is that in order to address the challenges of weak supervision, NeRd obtains program annotations from the QA pairs similar to the previous works, NumNet, NAQANet, NABERT, MTMSN and NMN, using
> - Various heuristic parsing of the templatized queries in DROP to get supervision of the reasoning type (max/min, diff/sum, count, negate).
> - Exhaustive search over all possible discrete operations to get supervision of the arguments in the reasoning
>
> Such heuristic supervision makes the learning problem significantly simpler for these models in the following ways
> - Enjoy specialized program having explicit information of the type of reasoning to apply e.g.,SUM(10,12)
> - A simplistic (contextual BERT-like)reader model to read query related information from the passage trained with direct supervision of the query span arguments at each step of the program
> - A programmer model that can be directly trained to decode the specialized programs
> - Executing numerical functions (e.g., difference, count, max, min) by performing the actual discrete operation as a post processing step on the model’s predicted program. For each of these previous works, it is possible to directly apply the learning objective on the space of decoded program, without having to deal with the discrete answer or any non-differentiability. Because of this dependency, NeRd’s architecture cannot support learning in absence of supervision of annotated programs.
>
> Note: NMN, similar to NeRd, creates the program annotation and also uses synthetic data augmentation techniques to learn functions like Count.
>
> Such heuristic techniques of program annotation or exhaustive search are not practical as the language of questions or the space of discrete operations become more complex. Also, our experiments on model performance bucketed by total number of passage entities, in Fig 5c) show how NMN’s performance drops exponentially with increasing number of entities because of its direct dependency on the exhaustive pre-computation of all possible outputs whereas WNSMN remains unaffected.
>
> Hence WNSMN takes a different approach to solve the cold-start problem without any additional annotation through
> - A noisy symbolic query decomposition that is oblivious to the reasoning type and simply based on generic parsing techniques
> - Our entity specific cross attention model extracting passage information relevant to each step of the decomposed query and learning an attention distribution over the entities of each type
> - Learning to apply discrete reasoning by employing neural modules that learn to sample the operation and the entity arguments
> - Leveraging a combination of neural and discrete modules when executing the discrete operation, instead of using only neural modules which need strong supervision of the programs for learning the functionality
> - Fundamentally different learning strategy by incorporating inductive bias through auxiliary losses and Iterative Maximal Likelihood for a more conservative initialization followed byREINFORCE.
>
> Note1: Some of the complexity of modeling arises from the challenging nature of learning only from the distance supervision of the answer alone.
>
> Note2: While WNSMN does not use any data augmentation techniques or pre-training of the reasoning modules, this is certainly one of the future works we consider (as mentioned in Sec. 4), to bridge the performance gap with the SoTA models.
>
> ---------------------------------------------------------------------------------------------------------------------------------------------
>
> Question: Why NeRd is not comparable to WNSMN?
>
> Response: We argue that NeRd is very similar to the other reasoning-free hybrid models (NumNet, NAQANet, NABERT, MTMSN) in terms of architecture and learning framework. Similar to them, NeRd also cannot learn in absence of any heuristic program annotation, hence cannot be compared with WNSMN which learns the reasoning path to the answer. Instead of learning to reason given the final answer supervision, they reduce the task to learning to decode the program, based on heuristic program annotation. NMN is the only reasoning based model that can support learning without any additional supervision, similar to us.
>
> To the best of our knowledge, WNSMN is the first work on modular networks (for fuzzy reasoning over text in RC framework) to handle the weak supervised setting and address the cold start problem without any specialized supervision of heuristic programs.

---

> > ### Author Response · Authors · 2020-11-18
> > **Addressing remaining questions (on NMN and WNSMN)**
> >
> > Question: Poor performance of NMN in DROP-num because of excluded question types by the original NMN paper (Gupta et al. (2020))?
> >
> > Response: NMN supports the same discrete space of operations (difference of numbers and dates, sum, max/min and count) as our model. However, in Gupta et al. (2020), the authors created a subset of DROP, based on various regular expression matching to the query. Our analysis showed that their full subsets can be covered by 19 regular expression sub-categories. Questions in each sub-category only differin stopwords or variations (e.g.,‘touchdown’ vs ‘field goal’) which do not affect the reasoning task.
> >
> > However, NMN’s architecture can still use weak supervision from the answer alone over all the training instances, making it a comparable setting with ours.
> >
> > ---------------------------------------------------------------------------------------------------------------------------------------------
> >
> > Question: Ablations of NMN not fair comparison
> >
> > Response: The ablations considered for NMN are in fact additional strong supervision signals used during training i.e. stepwise supervision of the program operator and arguments related to query segments and numerical/dates. On the other hand, WNSMN does not use any of these kinds of supervision, hence the fair comparison is with NMN when none of these additional supervisions is present.
> >
> > ---------------------------------------------------------------------------------------------------------------------------------------------
> >
> > Question: Would this dependency-based formalism work well elsewhere, or does it just happen to work well for DROP?
> >
> > Response: Even in DROP, dependency parsing can be noisy. Our motivation was to develop a reasoning framework which can sample the discrete operation and arguments even based on the noisy query decomposition. Note that these discrete sampling networks take as input not only the noisy program structure but also the original query. The model can implicitly learn whether (or to what extent) to depend on the full query or the query decomposition for different instances. Similarly the argument sampling networks learn how to manipulate the attention over entities for each step of the noisy query decomposition, to sample the correct entities for the discrete reasoning.
> >
> > ---------------------------------------------------------------------------------------------------------------------------------------------
> >
> > Question: Qualitative Analysis of WNSMN predictions
> >
> > Response: A manual analysis of WNSMN predictions of 334 instances of DROP-num Test questions where the gold answer is obtained by multiple (>1) actions sampled by WNSMN (including the top-1 action)is added in Appendix section A.3 (section on ‘Qualitative Inspection of WNSMN predictions’). Only 28 instances were found to have spurious action in top-1.

---

> > ### Comment · AnonReviewer3 · 2020-11-23
> > **NeRd**
> >
> > Thanks for the response. I think I understand your position a bit better now.
> >
> > However, I'm still not really convinced. In particular, on page 6, you say your training relies on the following:
> >
> > "With the assumption that the sampled actions are extensive enough to contain the gold answer..."
> >
> > You say above about prior methods that:
> >
> > "Such heuristic techniques of program annotation or exhaustive search are not practical as the language of questions or the space of discrete operations become more complex. Also, our experiments on model performance bucketed by total number of passage entities, in Fig 5c) show how NMN’s performance drops exponentially with increasing number of entities because of its direct dependency on the exhaustive pre-computation of all possible outputs whereas WNSMN remains unaffected."
> >
> > But assuming your sampled actions can enumerate reasonable programs and get the correct answer / get reward seems like a similar assumption about the "space of discrete operations" to me. I acknowledge the results in 5c, but given that this is entangled with other differences between the models, I don't really feel like it strongly substantiates this claim for me.
> >
> > I think the idea of anchoring to dependencies is a big advantage of your method. I like this. But it's a bit too hard to see a concrete benefit of this modeling nugget with experiments on a single dataset. Most of the arguments here are about scaling up, complexity when rolling out to new settings, etc. and when we're comparing a set of handcrafted systems around a single dataset, these arguments end up being largely theoretical.
> >
> > However, I will reflect on the rebuttal more closely and discuss with the other reviewers and area chair.

---

> > > ### Author Response · Authors · 2020-11-24
> > > **Addressing the above questions/comments**
> > >
> > > Thank you for your response. Please find our's below.
> > >
> > > Question: Assumption that the sampled actions are extensive enough to contain the gold answer
> > >
> > > Response: For the Hard-EM and REINFORCE to work, atleast one of the sampled actions need to have positive reward in atleast some of the training instances. This is the only assumption that any RL algorithm requires for training. It does not need any heuristic way of pruning which of the actions are spurious. In contrast, the prior work of NMN applies various heuristic pattern matching over the query to annotate the gold program.
> > >
> > > In the case of DROP-num, we attain 90% recall over all the sampled actions over Train (and 84% recall over the Test), which alludes towards the possiblity of using RL or Hard-EM algorithms.
> > >
> > > ----------------
> > >
> > > Question: Showcasing on a single dataset
> > >
> > > Response: Yes we agree that because of the challenges of the weak supervised setting, and being the first work (to our knowledge) to explore the framework with noisy programs and RL based training, we have focused only on a single dataset.

---

### Official Review · AnonReviewer4 · 2020-10-28
**Elaborate modeling with cutting-edge techniques, but limited on generalizability and lacks analysis.**

**Rating:** 7
**Confidence:** 3

**Review:**

ICLR 2021 Review

Title: Elaborate modeling with cutting-edge techniques, but limited on generalizability and lacks analysis.

This paper introduces a method to train neuro-symbolic module networks with weak supervision signal, and out-performs previous models on datasets containing numerical problems. Dislike previous works, WNSMN (Weakly-Supervised Neuro-Symbolic Module Network) doesn't rely on strong supervision and annotations but incorporates many inductive biases for numerical operations. It learns to execute programs that are extracted from dependency parse with noisy heuristics and are trained end-to-end in a reinforcement learning framework.

Neural Module Networks are a very promising trend for machine reading comprehension for that it explicitly involves the reasoning process, but training such a system requires expensive annotations. Weakly supervised learning was infeasible because of the exponentially large action space, but this paper manages to address this issue by some inductive biases. To make the searching efficient, WNSMN designs modules for the interaction between program and passage, as well as programs and entities. These inductive biases are implemented with an attention mechanism and make the action size tractable. Besides, the sampling process is also learnable, with two carefully designed samplers -- operator and argument samplers.

The training technique also plays an important role. Instead of training with discrete binary rewards and from scratch, WNSMN starts with hard EM objective with heuristic silver answers. The training process with the reinforcement learning objective adopts a multi-arm bandit assumption and entropy penalty. These techniques alleviate the problems of instability and overfitting.

Overall this paper is brilliant, adopts proper techniques, and elaborately designs the architecture. However, it's also limited in generalizability and needs more analysis.

Compared to the two baseline works, i.e. Neural Module Networks (NMNs) and GenBERT, WNSMN only works for numerical problems. The modules designed in the model, like the entity attention and operator sampler, are ad-hoc for problems involving dates and numbers. It's hard to extend WNSMN to more general cases. So it's questionable whether this framework could be applied to other problems of DROP like PERSON or VERB PHRASE.

One of the crucial pathologies of weakly supervised learning is the spurious prediction, where the model outputs the correct with wrong steps. However, this paper doesn't analyze this problem with WNSMN. So I'm wondering what percentage of the correct predictions of WNSMN is derived from the correct reasoning process? This could be studied with a small subset of human-annotated data and is helpful to rule out the possibility that the model learns some un-interpretable patterns.

The first step of WNSMN, i.e. query parser, is rule-based and makes less sense in contrast to other parts. It severely limits the ability of the model by just incorporating the first layer of the dependency parse tree. How can this model deal with complex patterns that are not step-by-step? How do you guarantee that the dependency relationship between steps is from left to right? I'm not convinced by the explanation "the QA model needs to be robust to such noise and additionally rely on the full query representation in order to predict the discrete operation." It will be clear if you can post statistics on the percentage covered by this simple heuristic.

Questions:

1. Is there a specific reason that you focus on numerical problems?
2. During hard EM, you choose the best action based on your current policy. How many good actions do you have in the first step? If there are too many good actions, it's still hard for the model to learn from correct signals.
3. Continued from question 2, why do you choose the best action, instead of smoothed over all the good actions?
4. To train the in-house baseline models, do you train them on the whole dataset or just the subset that involves numbers and dates? Is that possible that NMNs or GenBERT could learn useful patterns from other types of questions to answer numerical questions?

Presentation suggestions. The paper presentation is clear and dense overall. My suggestions would be:
1. Limit the content of describing the interactions. Section 2.1.1 is overwhelming and intimidating to me, with a lot of definitions and equations. The idea behind the interactions, however, is not that complex. Some descriptions of the methods could either be abridged or moved to the appendices, like the multi sliding windows and multi-scaling tricks given they're not your contribution. Also, the interactions between program/passage and program/entity are very similar, and you might be able to refactor these two paragraphs.
2. Curly brackets should be used for the set instead of ordered vectors. E.g. stacked attention windows in section 2.1.1.
3. Words in equations should be surrounded by \mathrm or \mathit to make them less like the production of many variables.

---

> ### Author Response · Authors · 2020-11-18
> **Addressing Questions on Generalizability and Qualitative Analysis**
>
>
> We thank the Reviewer for the detailed feedback and observations. In the following, we address the questions/comments raised.
>
> -----------------------------------------------------------------
>
> Question: Qualitative Analysis of WNSMN predictions
>
> Response: A manual analysis of WNSMN predictions of 334 instances of DROP-num test questions where the gold answer is obtained by multiple (>1) actions sampled by WNSMN (including the top-1 action) is added in Appendix section A.3 (section on ‘Qualitative Inspection of WNSMN predictions’). Only 28 instances were found to have spurious action in top-1.
>
> -----------------------------------------------
>
> Question: How is the model robust over dependency parsing?
>
> Response: The motivation of this work is to use noisy query decomposition as a program and learn the nature of discrete reasoning by predicting the discrete operation and the discrete arguments. Note that these discrete sampling networks take as input not only the noisy program structure but also the original query. The model can implicitly learn whether (or to what extent) to depend on the full query or the query decomposition for different instances. Similarly the argument sampling networks learn how to manipulate the attention over entities for each step of the noisy query decomposition, to sample the correct entities for the discrete reasoning.
>
> In order to make learning tractable in our weak supervised setting, we take this approach of combining both neural and symbolic tools in processing the query, but make provisions in the former to also directly learn from the original input since the latter can be noisy. Our experiments on the DROP-num dataset show this process is quite robust.
>
> -----------------------------------------------
>
> Question: Reason for focusing on numerical problems?
>
> Response: Since in this work we are investigating how neural models can incorporate explicit reasoning especially in the challenging weak supervised setting, we focus on only answering questions having numerical answer. We believe the effect of explicit reasoning is more directly observable for such questions.
>
> This is backed up by the category-wise performance comparison of reasoning-free language model GenBERT (reported in Geva et al. (2020)) with other hybrid models (MTMSN andNABERT+) that exploit numerical computation required in answering DROP questions. While, onDROP-num, there is an accuracy gap of 33% between the GenBERT model and the hybrid models (when all are trained on DROP only), there is only a 2-3% performance gap on the subset having answers as single span, despite the latter also needing reasoning. This evinces that the performance gap is indeed due to exploiting explicit reasoning under such strong supervised settings.
>
> -----------------------------------------------
>
> Question: Is it generalizable to other problems of DROP like PERSON or VERB PHRASE?
>
> Response: In order to work on other kinds of entities like PERSON or NOUN PHRASE, only the first preprocess-ing step of extracting relevant entities from the passage would differ. Instead of identifying numbers and dates, it would need a way to symbolically define entities (can be numbers, dates or named entities like PERSON) and link their mentions in the passage to canonicalized entity names.
>
> Once separate entity-lists along with their mention locations have been created for each required entity-type,the same Entity-Specific Cross Attention for Information Extraction and Discrete Reasoning steps described in Sec 2.1 can be followed to reach the answer.
>
> As long as the desired discrete operations can be defined on other named entity types and have an output space comparable to the final answer (e.g.,space of numerical quantities) which allows us to define some reward function, the same architecture can be used on other entity types.
>
> ----------------------------------------------
>
> Question: Why do you choose the best action, instead of smoothed over all the good actions?
>
> Response: We had done additional experiments in the Iterative Maximal Likelihood Objective where instead of taking the “most likely” good action according to the current policy as the best one, we take a convex combination of the most likely and the least likely action according to the current policy. This is described in details in Appendix Section A.6 (in paragraph titled ‘Iterative ML Objective’). Since this only led to marginal improvement in training, for simplicity we do not report it in the main paper. Further extending this to learning over all good actions is essentially what the REINFORCE objective aims at.
>
> -----------------------------------------------
>
> Question: How many good actions are there?
>
> Response: Empirical results show avg. number of good actions (where there is more than 1 good action) is 2.25 (on DROP-num Test). This low number alludes towards the possibility of Iterative ML based hard-EM techniques being a reasonable conservation initialization for REINFORCE.

---

> > ### Author Response · Authors · 2020-11-18
> > **Addressing Questions on other Baselines**
> >
> > Question: What is the effect of training the baselines on full DROP (not restricted to numerical answers)
> >
> > Response:
> > - GenBERT models (vanila GenBERT and GenBERT+ND+TD) performance improve by 3.5-3.89%on DROP, when training is not restricted to numerical answers subset.
> > - The other GenBERT models (GenBERT + ND, GenBERT + TD) give minor improvement of 0.24-0.96%. Hence in our empirical comparison in Table 3 we report GenBERT instead of GenBERT-num.
> > - On the other hand, NMN’s performance degrades by 1.57% when trained on full DROP (without restricting to numerical answers subset).
> > - However in the case of Pruned-DROP, training on all answer types gives a 2.6% improvement.

---

### Official Review · AnonReviewer2 · 2020-10-28
**interesting model, but there are positioning and presentation issues**

**Rating:** 5
**Confidence:** 3

**Review:**


The paper proposes a new model for numerical reasoning in machine comprehension. Given a passage and a query, the model outputs an arithmetic expression over numbers/dates in the passage (e.g. max(23, 26, 42)). The model is trained with weak supervision in the form of numerical answers only. This weak supervision is used to define reward for reinforcement learning training. A key claimed advantage of the model compared to the prior art is that it trains end-to-end from the rewards as the only form of supervision. This is contrasted to  neural module networks, which require program supervision for good performance, as well as GenBERT, which requires additional synthetic training data for pretraining. Two key quantitative results include:
better performance on the DROP-num datasets, compared to NMNs with less supervision and GenBERT without data augmentation
comparable to strongly-supevised NMN performance on DROP-Pruned-num.

The general approach is quite elegant and makes sense. It is encouraging that the paper reports successful training with RL. It is also important to build models that use less extra supervision.

That said, I have some concerns regarding the paper’s positioning. The introduction, as well as many other places in the text categorizes some of the prior art as “learning a multi-type answer predictor over different reasoning types (e.g., max/min, diff/sum, count, negate) and directly predicting the corresponding numerical expression, instead of learning to reason”. What is learning to reason then? For example, in NAQANet the model predicts whether the numbers in the passage should be summed or subtracted from each other, why is this not learning to reason? Second, calling this model a “module network” is misleading, in my opinion. In neural module networks modules learn to do things, and here the key modules of discrete reasoning are predefined. The model also contains “modules” that are conditioned on different question spans. But there is no experiment checking whether having multiple such modules is actually useful. There is furthermore no qualitative explanation of what these modules are supposed to do and what they are actually doing. Lastly, I am not sure the use of synthetic data for GenBERT can be called “strong supervision”. Data augmentation and strong supervision are not the same thing.

The paper is very dense and is quite hard to read. A lot of space is allocated to a very detailed technical presentation of “modelling interactions”, while the high-level picture of how the model functions is still hard to grasp. For example, Figure 2 is confusing because it has a “Stacked Span Prediction” pathway that leads to nowhere. Reading the text I find that apparently the output of this part is actually used for “modelling interaction between programs and number entities”, which is in the right part of the figure. These basic high-level architectural decisions are hard to understand as the reader is overwhelmed by technical details, such as sliding windows and scaling factors for various attentions.

The way the results are displayed in the table is somewhat confusing. The fully-supervised NMN baseline is shown in Table 2 for DROP-Pruned-num but is not shown in Table 1 for Table-num. Instead, it can be found in Table 3. I would recommend presenting all results in one table, even if it shows that the current model performs worse than others with more supervision or data augmentation. Furthermore, I think that comparing top-k accuracies for the proposed model and top-1 accuracies for other models, as it is done in Table 3, does not make sense.

In summary, while I think the paper might be proposing an interesting model with promising results, I also think that presentation needs work. It would be great to see a clearer high-level explanation of the difference between the proposed model and the prior work. To this end, the prior work should be better discussed (notably there is no Related Work section at the moment). Besides, more quantitative and/or qualitative results are needed to support the hypothesis that the model performs “noisy query execution”.

Other minor comments:
- it is the first time I see the word skyline used to mean “the baseline from above”
- there is a lot of really long sentences in the paper, which makes the reading very hard. I’d recommend to try and break them up into shorter ones.

---

> ### Author Response · Authors · 2020-11-18
> **Addressing Positioning Issue and clarifying 'Learning to Reason'**
>
> We thank the Reviewer for the detailed feedback and observations. In the following, we address the questions/comments raised.
>
> -----------------------------------------------------------------------------------------------------------------------------------------------------------------
>
> Question: Positioning Issue: What is learning to reason then? For example, in NAQANet the model predicts whether the numbers in the passage should be summed or subtracted from each other, why is this not learning to reason?
>
> Response: Each of these previous models (NumNet, NAQANet, NABERT+, MTMSN) heuristically identifies the kind of reasoning type involved (max/min, diff/sum, count, negate, etc), and based on that, it performs an exhaustive search to obtain supervision of the numerical expression that leads to the correct answer e.g.,max(23, 26, 42) for the answer 42. Because of the templatized nature ofDROP questions, this heuristically obtained supervision is often quite accurate. Hence these models frame the QA task as a classification task or a supervised decoding task, where, having extracted all the numerical entities, they learn to predict/decode the correct numerical expression, without having to deal with the final answer at all. Their model architecture does not allow them to learn in absence of this supervision.
>
> The problem of Learning to Reason, on the other hand, needs a weak supervised setting where the exact supervision of the reasoning is not present and the model learns the reasoning path to reach to the answer through the distant supervision based on the end task. Prior works on learning to reason (Hu et al., 2017; Liang et al., 2017; Guo et al., 2017; Santoro et al., 2017) have focused on simpler tasks like QA on structured Knowledge Bases or Visual Question Answering on synthetic image datasets like CLEVR. The original work which introduced and conceptualized module networks,(Hu et al., 2017) had a similar weak supervised motivation. However, in the numerical reasoning based RC framework, NMN is the only reasoning based model that also allows learning in the weak supervised setting, making it a fair baseline for us.
>
> Related References:
>
> 1. Hu et. al, Learning to reason: End-to-end module networks for visual question answering, ICCV2017
> 2. Guo et. al, Learning to query, reason, and answer questions on ambiguous texts, ICLR 2017
> 3. Santoro et. al, A simple neural network module for relational reasoning, NIPS 2017
> 4. Liang et. al, Neural symbolic machines: Learning semantic parsers on freebase with weak supervision, ACL 2017

---

> > ### Author Response · Authors · 2020-11-18
> > **Addressing Remaining Comments (on other baselines and qualitative analysis)**
> >
> > Question: Qualitative Analysis of WNSMN predictions
> >
> > Response: A manual analysis of WNSMN predictions of 334 instances of DROP-num test questions where the gold answer is obtained by multiple (>1) actions sampled by WNSMN (including the top-1 action) has been added in Appendix section A.3 (section on ‘Qualitative Inspection of WNSMN predictions’). Only 28 instances were found to have spurious action in top-1.
> >
> > ----------------------------------------------------------------------------------------------------------------------------------
> >
> > Question: Use of synthetic data for GenBERT cannot be called “strong supervision”
> >
> > Response: Good comment! The synthetic data used in GenBERT has additional supervision of the gold numerical expression that leads to the answer. However, the most striking aspect of the synthetic data is the similarity with the DROP dataset in terms of the domains (nfl and history) of the passages and vocabulary and the distribution of the range of numerical quantities. In Appendix Section A.4.2 (PreTraining Data for GenBERT), we have elaborated on this analysis, based on which we feel the pre-training data is not generic enough and may have been tailored to the specific target dataset. Further because of the templatized nature of DROP questions, the synthetic data is even more similar to DROP and hence effective.
> >
> > ----------------------------------------------------------------------------------------------------------------------------------
> >
> > Question: Lack of Related Work
> >
> > Response: We have added a section (Section 5) distinguishing WSNMN from the two closest genre of models i.e. reasoning free hybrid models (including a missed reference pointed out by Reviewer 3) and reasoning based NMN model.
> >
> > ----------------------------------------------------------------------------------------------------------------------------------
> >
> > Question: Calling this model a “module network” is misleading, in my opinion
> >
> > Response: We agree that term module network have been used for purely neural modules which learn the functionality from scratch However WNSMN leverages a combination of neural components (fore.g.,to sample operator or argument or number of arguments to sample) and discrete operations (fore.g.,difference, sum ). This allows us to
> > i) keep the learning tractable in our weak supervised setting
> > ii) potentially scale to arbitrarily complex non-linear operations (e.g.,log, cos) for which the modeling complexity can exponentially increase, especially in absence of strong supervision. Accordingly we named our model “Neuro-Symbolic Module Network” but we do not have particular reservations regarding the reviewer’s opinion on this.
> >
> > ----------------------------------------------------------------------------------------------------------------------------------
> >
> > Question: Comparing top-K results of WNSMN with Top-1 results of other strong supervised models in Table 3
> >
> > Response: We do not intend to compare the top-K results of WNSMN with the top-1 results of the strong supervised models but instead quantize how far the weak supervised models are or where they rank in comparison to the strong supervised SoTA models. The promising recall over the top-K actions suggests that more sophisticated RL algorithms with better exploration strategies can possibly bridge this performance gap.

---

### Decision · Program_Chairs · 2021-01-07
**Final Decision**

**Decision:**

Reject

**Comment:**

This paper proposes a weakly supervised model for numerical reasoning. After discussion with the reviewers it seems that it is already known that training NMNs directly on DROP is not successful and requires taking additional measures. Past work (NERD) has resorted to using data augmentation, and this work encodes it directly to the model. This paper needs to show the advantages of their approach and that it generalizes better to other scenarios. Other minor issues include (a) clarity fo writing (b) focus on a subset of questions (c) no evaluation on other numerical datasets (d) mild inaccuracies w.r.t prior work (GenBERT)